# Arabinose Plays an Important Role in Regulating the Growth and Sporulation of *Bacillus subtilis* NCD-2

**DOI:** 10.3390/ijms242417472

**Published:** 2023-12-14

**Authors:** Yifan Fu, Xiaomeng Liu, Zhenhe Su, Peipei Wang, Qinggang Guo, Ping Ma

**Affiliations:** 1College of Plant Protection, Agricultural University of Hebei, Baoding 071000, China; 15230233219@163.com; 2Key Laboratory of IPM on Crops in Northern Region of North China, Integrated Pest Management Innovation Centre of Hebei Province, Institute of Plant Protection, Hebei Academy of Agriculture and Forestry Sciences, Ministry of Agriculture and Rural Affairs of China, Baoding 071000, China; 13315286401@126.com (X.L.); suzhenhe0202@126.com (Z.S.); wangpeipei0010@163.com (P.W.)

**Keywords:** Phenotype MicroArray, *Bacillus subtilis*, L-arabinose, sporulation, transcriptome

## Abstract

A microbial fungicide developed from *Bacillus subtilis* NCD-2 has been registered for suppressing verticillium wilt in crops in China. Spores are the main ingredient of this fungicide and play a crucial role in suppressing plant disease. Therefore, increasing the number of spores of strain NCD-2 during fermentation is important for reducing the cost of the fungicide. In this study, five kinds of carbon sources were found to promote the metabolism of strain NCD-2 revealed via Biolog Phenotype MicroArray (PM) technology. L-arabinose showed the strongest ability to promote the growth and sporulation of strain NCD-2. L-arabinose increased the bacterial concentration and the sporulation efficiency of strain NCD-2 by 2.04 times and 1.99 times compared with D-glucose, respectively. Moreover, L-arabinose significantly decreased the autolysis of strain NCD-2. Genes associated with arabinose metabolism, sporulation, spore resistance to heat, and spore coat formation were significantly up-regulated, and genes associated with sporulation-delaying protein were significantly down-regulated under L-arabinose treatment. The deletion of *msmX*, which is involved in arabinose transport in the *Bacillus* genus, decreased growth and sporulation by 53.71% and 86.46% compared with wild-type strain NCD-2, respectively. Complementing the mutant strain by importing an intact *msmX* gene restored the strain’s growth and sporulation.

## 1. Introduction

Plant soil-borne diseases such as verticillium wilt and fusarium wilt cause significant losses in plant production and are very difficult to control. Microbial fungicides using living microorganisms as active compounds are effective and environmentally friendly methods to suppress plant soil-borne diseases and reduce the amount of chemical fungicides needed [1,2]. The control capabilities of biocontrol agents are influenced by their concentration in the plant rhizosphere. Therefore, it is important to increase the application dose of biocontrol agents.

*B. subtilis* is an important resource for developing microbial fungicides due to its ability to produce a variety of antibiotics and form highly resistant spores [3]. Wettable powders and dry powder seed-coating agents are the main formulations of microbial fungicides for suppressing plant soil-borne diseases. However, processing these two formulations involves an instantaneous high temperature of as high as 170 °C to dry the bacteria, and only the spores can survive under such high temperatures. Therefore, microbial fermentation should consider the bacterial concentration as well as the sporulation [4]. An ideal fermentation system first increases the fermentation level of the bacterium and, subsequently, carries out the maximum possible conversion of the bacteria into spores via nutrient regulation and other methods [5].

The carbon and nitrogen sources in a medium are the main factors affecting bacterial growth and sporulation [6,7,8]. Suitable and sufficient carbon and nitrogen sources can promote the growth of the bacterium, but sporulation generally occurs in unfavorable environments, such as nutrient starvation [9]. During the stable phase of bacterial growth, residual carbon and nitrogen sources in the medium might inhibit the sporulation of *Bacillus* spp. [10]. In microbes, carbohydrates are catabolized into pyruvate, which enters the tricarboxylic acid cycle, mainly through the glycolytic pathway (EMP) and pentose phosphate pathway (PPP). By comparing the effects of the intermediate products in the EMP and PPP pathways on the sporulation of *B. subtilis*, it was found that the sugars existing in the PPP pathway but not in the EMP pathway could increase the sporulation of *Bacillus*. Therefore, it was concluded that the PPP pathway is an important carbohydrate catabolic pathway that affects sporulation [11]. In addition, metal ions such as Ca^2+^, Mg^2+^, Zn^2+^, and Mn^2+^ also affect the sporulation of *B. subtilis*, which is promoted via the addition of appropriate concentrations of the metal ions in the medium [12,13,14]. Different strains have different nutrient requirements suitable for their growth and sporulation; therefore, specific nutrients must be explored for a specific strain.

*Bacillus subtilis* NCD-2 showed a promising biocontrol effect against plant soil-borne diseases and was developed as a commercial microbial fungicide against cotton verticillium wilt in China [15,16,17]. This study aimed to screen the nutrients suitable for the growth and sporulation of strain NCD-2 and then explore the mechanism for regulating sporulation via these nutrients. The results of this study will provide important information for the large-scale and efficient fermentation of strain NCD-2.

## 2. Results

### 2.1. Screening of Nutrients That Facilitate the Metabolism of Strain NCD-2

The metabolic activities of strain NCD-2 for carbon, nitrogen, phosphorus, sulfur, and trace elements were determined using Biolog Phenotype MicroArray (PM) technology (Appendix A). Regarding carbon source utilization, strain NCD-2 had higher metabolic activity under L-arabinose, D-arabinose, D-xylose, D-ribose, and D-glucosamine treatment. Regarding nitrogen source utilization, strain NCD-2 had higher metabolic activity under cysteine treatment. Regarding phosphorus and sulfur source utilization, strain NCD-2 had low metabolic activities in 59 phosphorus sources and 35 sulfur sources.

### 2.2. Effects of L-Arabinose, D-Ribose, and D-Xylose on Growth and Sporulation Efficiency of Strain NCD-2

The effects of different carbohydrates on the growth of strain NCD-2 were evaluated (Figure 1a). When D-glucose was used as a carbon source, the bacterial concentration of strain NCD-2 reached a maximum of 2.55 × 10^7^ CFU/mL, while that using L-arabinose was 2.04 times higher, at 5.20 × 10^7^ CFU/mL. When D-ribose and D-xylose were used as sole carbon sources, the bacterial concentrations of strain NCD-2 reached maximums of 3.90 × 10^7^ CFU/mL and 3.10 × 10^7^ CFU/mL, respectively. It was revealed that L-arabinose was the most suitable carbon source to promote the growth of strain NCD-2.

The effects of different carbohydrates on the sporulation of strain NCD-2 were evaluated (Figure 1b). Using D-glucose, L-arabinose, D-ribose, and D-xylose as sole carbon sources, the sporulation efficiencies of strain NCD-2 were 2.33%, 27.12%, 11.40%, and 15.39% after 24 h of inoculation, respectively. The sporulation efficiencies of strain NCD-2 were 41.35%, 82.18%, 20.26%, and 20.45% after 48 h of inoculation when using D-glucose, L-arabinose, D-ribose, and D-xylose as carbon sources, respectively. The sporulation efficiencies of strain NCD-2 increased from 24 h to 48 h of inoculation. The increase in sporulation with L-arabinose was also confirmed using microscopy (Appendix A). The results indicate that L-arabinose increased the sporulation of strain NCD-2 during the early and later growth stages, but D-ribose and D-xylose only increased the sporulation of strain NCD-2 in the early growth stage compared with D-glucose.

### 2.3. Effects of Different Proportions of L-Arabinose and D-Glucose on Sporulation of Strain NCD-2

The effects of different proportions of L-arabinose and D-glucose on the sporulation of strain NCD-2 were evaluated (Figure 2). Forty-eight hours after inoculation, the sporulation efficiencies of strain NCD-2 were 41.35% and 82.43% when using D-glucose and L-arabinose as sole carbon sources, respectively. The sporulation efficiency of strain NCD-2 gradually decreased with the decrease in the L-arabinose proportion in a mixture of L-arabinose and D-glucose. The sporulation efficiencies of strain NCD-2 were 68.82%, 56.62%, and 32.79% when L-arabinose and D-glucose were at proportions of 2:1, 1:1, and 1:2, respectively.

### 2.4. Transcriptome Analysis

The effects of L-arabinose and D-glucose on the gene expression of strain NCD-2 were compared using transcriptome sequencing. Compared with D-glucose, a total of 1483, 1773, and 2271 differential expression genes (DEGs) were identified under L-arabinose treatment at 8 h, 12 h, and 16 h after inoculation, respectively (Figure 3). GO annotations revealed that the DEGs associated with sporulation (GO:0043934), spore walls (GO:0031160), endospore-forming forespores (GO:0042601), and asexual sporulation (GO:0030436) were significantly up-regulated at 8 h and 12 h post-inoculation. However, only the DEGs associated with spore germination (GO:0009847) were enriched at 16 h post-inoculation (Figure 4). KEGG enrichment analysis found that ABC transporters (map02010), polyketide sugar unit biosynthesis (map00523), and ribosomes (map03010) were significantly enriched at 8 h and 12 h (Figure 5).

### 2.5. Confirmation of Transcriptional Results with qRT-PCR

To verify the transcriptome results, 14 DEGs associated with arabinose transportation, sporulation, spore resistance to heat, etc., were selected to analyze the expression under L-arabinose treatment with qRT-PCR. The results reveal that all 14 genes showed consistent expression trends with transcriptome analysis, indicating the transcriptome results were reliable and could be used for further experimental analysis (Appendix A).

### 2.6. Analysis of Genes Associated with Sporulation in Strain NCD-2

The Venn plot shows that 717 genes were significantly differentially expressed at all three time points (Appendix A), and most of the up-regulated genes were involved in sporulation (Table 1). Among them, *sigK* was a transcriptional regulator-encoding spore formation. *cotE*, *cotF*, *cotG*, *cotS*, *cotT*, *cotV*, *cotW*, *cotX*, and *yheD* were responsible for encoding spore coat proteins. *dpaA* and *dpaB* were responsible for synthetases-encoding pyridine dicarboxylic acid (DPA), a substance within the spore core. *gerBA*, *gerE*, *gerQ*, and *gerT* were responsible for encoding spore germination proteins. *spoIIIAH* and *spoIIQ* encoded polymeric complexes that connected the forespore and mother cell [18]. *spoIVB* and *spoIVFB* encoded the activating proteins of protease and Sig-K, which catalyzed the formation of the spore cortex, respectively. *sspA*, *sspB*, *sspD*, and *sspE* encoded small acid-soluble proteins (SASPs) associated with spore resistance to heat. *sdpC* encoded a cannibalism factor that delayed sporulation in *B. subtilis* and was significantly down-regulated at all three time points.

### 2.7. Deletion of the msmX Gene Decreased the Sporulation Efficiency in Strain NCD-2

To confirm that L-arabinose influenced sporulation, the *msmX* gene, which encodes ATPase, responsible for arabinose uptake, was deleted from strain NCD-2 (Δ*msmX*). Additionally, an *msmX*-complemented strain was developed for the Δ*msmX* mutant (CP*msmX*). The growth and sporulation efficiencies of the WT, Δ*msmX*, and CP*msmX* strains were compared in M9 medium with L-arabinose as the carbon source (Figure 6a). The results show that the bacterial concentrations of WT were 4.60 × 10^7^ CFU/mL, 5.25 × 10^7^ CFU/mL, and 4.35 × 10^7^ CFU/mL at 24, 36, and 48 h post-inoculation, respectively. Comparatively, strain Δ*msmX* decreased growth, and the bacterial concentrations were 1.18 × 10^7^ CFU/mL, 2.25 × 10^7^ CFU/mL, and 2.43 × 10^7^ CFU/mL after 24 h, 36 h, and 48 h of inoculation, respectively. Meanwhile, the complemented strain (CP*msmX*) restored growth, and the bacterial concentrations were 3.53 × 10^7^ CFU/mL, 6.20 × 10^7^ CFU/mL, and 5.45 × 10^7^ CFU/mL after 24 h, 36 h, and 48 h of inoculation, respectively.

The sporulation efficiencies of wild-type strain NCD-2 (WT), *msmX*-null mutant (Δ*msmX*), and its complemented strain (CP*msmX*) were also compared in M9 medium with L-arabinose as the carbon source (Figure 6b). The results show that the sporulation efficiencies of strain WT were 27.12%, 70.95%, and 82.18% at 24, 36, and 48 h post-inoculation, respectively. Comparatively, strain Δ*msmX* decreased the sporulation efficiencies by 7.17%, 11.67%, and 11.13%, at 24, 36, and 48 h post-inoculation, respectively. Meanwhile, the complemented strain (CP*msmX*) increased the sporulation efficiencies by 44.63% and 65.44% at 36 h and 48 h post-inoculation, respectively. The regulation of strain NCD-2’s growth and sporulation by *msmX* was also confirmed via microscopic observation (Appendix A).

## 3. Discussion

Plant soil-borne diseases such as verticillium wilt and fusarium wilt are difficult to suppress with chemical fungicides, rotation, and resistant varieties, mainly due to phytopathogens that can survive for decades in soil [19]. Recent studies have revealed that applying microbial fungicides can successfully suppress plant soil-borne diseases, and the genus *Bacillus* is a major resource for developing microbial fungicides [20]. Spores of *Bacillus* species are resistant to stresses and are used as the key ingredient in the formulation of microbial fungicides. Generally speaking, the control effect of plant soil-borne diseases is positively correlated with the population of bacteria colonized in the plant rhizosphere [21]. To obtain an ideal biocontrol effect, the amount of spores applied to the soil should be increased as much as possible. Therefore, promoting the yield of spores during the fermentation process of *Bacillus* is important for reducing the cost and ensuring the wide application of micro-fungicides. Promoting the growth of bacteria and the yield of spores during the fermentation process is one of the key factors in reducing the cost of bio-fungicides. It is known that the sporulation process and final spore yield depend on carbohydrates and amino acids [22]. The combined effects of yeast extract, peptone, and glucose enhanced the spore yield of *B. megaterium* [23]. Likewise, the addition of glucose and ribose into the sporulation medium increased the spore yields of *B. subtilis* and *B. cereus* [6,11,24]. In this study, we focused on nutrients that promoted strain NCD-2’s growth and spore formation, which required screening for a large number of nutrients due to different strains having different nutrient requirements. Phenotype MicroArrays (Biolog) are commercially available microplate assays that can be used to test more than 1000 phenotypic traits simultaneously by recording the microorganism’s respiration over time on many distinct substrates [25,26]. Therefore, PMs can be used to screen nutrients suitable for the catabolism of a specific organism quickly and in high throughput, which has the advantages of producing a large amount of information and saving time [27]. In this way, the catabolic capability of strain NCD-2 in approximately 200 carbon sources, 400 nitrogen sources, and 100 phosphorous and sulfur sources was determined using PMs. The results show that strain NCD-2 had a higher metabolic capacity with L-arabinose, D-xylose, and D-ribose as sole carbon sources, among which L-arabinose significantly increased the bacterial concentration and sporulation efficiency of strain NCD-2.

*B. subtilis* could grow on a medium with L-arabinose as the sole carbon and energy source. Without L-arabinose, the AraR protein bounded to a site within the *araABDLMNPQ-abfA* operon promoter region, preventing transcription. With L-arabinose, a conformational change was induced in AraR, such that recognition and binding to DNA were no longer possible, and the operon could be expressed [28]. After entering the cell, L-arabinose was sequentially converted to L-ribulose, L-ribulose 5-phosphate, and D-xylulose 5-phosphate via the action of L-arabinose isomerase (encoded by *araA*), L-ribulokinase (encoded by *araB*), and L-ribulose-5-phosphate 4-epimerase (encoded by *araD*), respectively. D-xylulose 5-phosphate was further catabolized through the pentose phosphate pathway [29,30]. Transcriptome analysis showed that L-arabinose strongly up-regulated *araABDLMNPQ-abfA* operon expression in strain NCD-2 (accession number: SUB12858722), and genes associated with sporulation were also strongly up-regulated (Table 1). Moreover, compared with using L-arabinose as the sole carbon source, the sporulation efficiency of strain NCD-2 significantly decreased when both glucose and arabinose were present in the M9 medium (Figure 2), which might be because glucose repressed the expression of both *araE*, a gene for the L-arabinose transporter, and the *ara* operon at the transcriptional level [31,32]. Thus, L-arabinose might be involved in regulating the expression of genes related to sporulation in strain NCD-2 by regulating the *ara* operon.

Previous studies found that the AraNPQ-MsmX system was involved in the transport of arabinans, and knocking out *araNPQ* reduced the growth rate of *B. subtilis* [28,33]. Therefore, deleting the *msmX* gene, which encodes ATPase to provide energy to the AraNPQ transporter, inevitably led to a decrease in the growth rate of *B. subtilis*. In this study, *msmX* was deleted from wild-type strain NCD-2, and the mutant reduced the bacterial growth and sporulation efficiency of strain NCD-2 with L-arabinose as the sole carbon source (Figure 6)—which is consistent with the previous study—but not with D-glucose as the sole carbon source. L-arabinose entered the metabolic process of strain NCD-2 via the AraNPQ-MsmX system, affecting its growth and sporulation. These results provide knowledge for effectively improving growth and spore production during the fermentation of strain NCD-2.

Autolysis of *B. subtilis* was observed during fermentation, resulting in a large number of cell deaths and reducing the bacterial fermentation concentration [34]. Many factors led to the autolysis of the bacterium [35,36,37,38]. Among them, a phenomenon of “cannibalism” was described [39,40], in which the master regulator of sporulation Spo0A was active and released two toxins, Skf and SdpC, to kill Spo0A-inactive sister cells. The nutrients released by the dead cells were used for the growth of cells that were not yet committed to sporulating. In this study, it was observed that cell autolysis produced a large amount of cell debris in the medium with D-glucose as the sole carbon resource, but not in the medium with L-arabinose. In the transcriptome analysis, L-arabinose significantly down-regulated the transcription of *sdpC* compared with D-glucose at 8 h, 12 h, and 16 h post-inoculation. Therefore, we speculated that L-arabinose increased the bacterial concentration of strain NCD-2 by inhibiting the process of “cannibalism” in strain NCD-2.

## 4. Materials and Methods

### 4.1. Bacterial Strains and Growth Conditions

The strains used in this study are listed in Table 2. *B. subtilis* strains were stored at −80 °C in LB medium containing 30% glycerol. Strain CP*msmX* was inoculated in M9 medium containing chloramphenicol (Sangon Biotech, Shanghai, China) at a final concentration of 5 μg·mL^−1^ and cultured at 30 °C under continuous agitation (180 rpm).

### 4.2. Biolog Phenotype MicroArray Analysis

The metabolic phenotype of *B. subtilis* strain NCD-2 in 755 nutrients was evaluated using the Biolog Phenotype MicroArray system (Biolog, Hayward, CA, USA). Ninety-six-well PM1-8 MicroPlates^TM^ (Biolog, Hayward, CA, USA), including a carbon source (PM1 or PM2A), a nitrogen source (PM3B or PM6-8), and a phosphorus source and sulfur source (PM5), were assayed. The names of nutrients are described in the literature of Bochner et al. [25]. The experiment was conducted according to the procedures developed by the manufacturer [27]. Briefly, strain NCD-2 was grown overnight at 33 °C on BUG + B plates (Biolog, Hayward, CA, USA). A single colony was selected, grown on BUG + B plates again, under the same conditions. Cells were picked up from the plates with sterile cotton swabs and transferred into a sterile, capped tube containing 20 mL of inoculation fluid (IF-0a, Biolog, Hayward, CA, USA). Cell density was adjusted to 81% transmittance on the Biolog turbidimeter. The PM1-8 MicroPlates^TM^ were inoculated with the cell suspension (100 μL/well), and then incubated at 30 °C for 48 h in the OmniLog incubator (Biolog, Hayward, CA, USA). The plates were scanned every 15 min, and the results were analyzed and plotted using the OmniLog software (OL_FM 12 analysis package, v1.2) at the end of the incubation.

### 4.3. Determination of Cell Concentration and Sporulation Efficiency

Strain NCD-2 was inoculated in Luria–Bertani (LB) broth and cultured at 30 °C and 180 rpm for 12 h. Cells were collected via centrifugation at 10,000 rpm for 2 min and adjusted to OD_600_ = 1.0 with sterile water. The cell suspension was inoculated in 100 mL M9 medium (12.8 g·L^−1^ of Na_2_HPO_4_·7H_2_O, 3 g·L^−1^ of KH_2_PO_4_, 0.5 g·L^−1^ of NaCl, 1 g·L^−1^ of NH_4_Cl, 0.24 g·L^−1^ of MgSO_4_, 0.011 g·L^−1^ of CaCl_2_, 4 g·L^−1^ of D-glucose) at a 1% inoculation volume. To evaluate the effects of different carbohydrates on the sporulation of strain NCD-2, the D-glucose was replaced with the same concentration of L-arabinose, D-xylose, and D-ribose (Beijing Solarbio Science & Technology Co., Ltd., Beijing, China). Strain NCD-2 was continuously cultured for 24, 36, and 48 h at 30 °C, and the sporulation efficiency was then determined following the protocol described by Eswaramoorthy et al. [41].

### 4.4. RNA Extraction and RNA Sequencing

Strain NCD-2 was inoculated in 100 mL of M9 medium with D-glucose or L-arabinose as the sole carbon source and cultured at 30 °C and 180 rpm for 8 h, 12 h, and 16 h. Cells were harvested via centrifugation at 4 °C and 10,000 rpm for 5 min, and three biological replicates were included. The bacterium was rapidly frozen with liquid nitrogen and stored at −80 °C. The total RNA of the collected bacteria was extracted according to the instructions of the RNAprep Pure Cell/Bactria Kit (TianGen Biotech, Beijing, China), and the quality and concentration of total RNA were measured with the NanoDrop 2000 (Thermo Fisher Scientific, Waltham, MA, USA). The cDNA library construction and RNA sequencing (RNA-seq) were performed with the Illumina platform at Majorbio Co., Ltd. (Shanghai, China).

### 4.5. Transcriptome Data and Differential Gene Expression Analysis

The transcriptome raw data were uploaded to the NCBI-SRA database (accession number: SUB12858722), using fastp (https://github.com/OpenGene/fastp/ (accessed on 10 February 2023)) to remove low-quality reads and adapters from the data. Then, these clean reads were mapped onto the reference genome (*Bacillus subtilis* NCD-2) using Bowtie (http://bowtie-bio.sourceforge.net/index.shtml/ (accessed on 11 September 2021)). The screening criteria for differentially expressed genes (DEGs) were a |log_2_ (Fold Change)| of >1 and an adjusted *p*-value of <0.05. DEGs were used for Gene Ontology (GO) and Kyoto Encyclopedia of Genes and Genomes (KEGG) enrichment analyses using Goatools (https://github.com/tanghaibao/goatools/ (accessed on 20 September 2021)) and the “clusterprofile” R package, respectively. The enrichment results were filtered with the parameter of a *p*-value of <0.05.

### 4.6. Confirmation of Transcriptome Analysis Results

To validate the transcriptome results, 14 genes (Appendix A) were selected for the expression analysis with qRT-PCR. Primers were designed using Primer Premier 5.0 software (Applied Biosystems, Waltham, MA, USA). *B. subtilis* strain NCD-2 was cultured in M9 medium with L-arabinose or D-glucose as the sole carbon source and cultured at 30 °C under continuous agitation (180 rpm). Cells were collected via centrifugation at 10,000 rpm for 1 min after 8 h, 12 h, and 16 h of incubation, respectively. Total RNA was extracted as described above and adjusted to 50 ng·μL^−1^. The extracted total RNA was used as a template to synthesize the first-strand cDNA using TransScript^®^ One-Step gDNA Removal and cDNA Synthesis SuperMix (TransGen Biotech, Beijing, China), and all the cDNA sample concentrations were diluted 3-fold with double distilled water. qRT-PCR assays using SYBR Green as a detector were performed in a StepOne™ Real-Time PCR system, conditions for amplification were according to the instructions of TransStart^®^ Top Green qPCR SuperMix (TransGen Biotech, Beijing, China). Three replicates were set up for each gene detection. The *gyrB* gene was used as the internal reference gene, and the relative change in target gene expression was calculated using the formula 2^−ΔΔCt^ [42].

### 4.7. Function Analysis of msmX Gene

To delete the *msmX* gene from strain NCD-2, *msmX* upstream fragments were amplified using primers *msmX*-P1 (CGAGCTCTTTCAGCGGTTCGGGTG) and *msmX*-P2 (GGGGTACCGATCAAAAAAACCGGACATGGGG), and *msmX* downstream fragments were amplified using primers *msmX*-P3 (GGGGTACCACCCAGCCATCTAACATCCCCC) and *msmX*-P4 (GCTCTAGATCCCGGTTCGATTGTGTCTG). The upstream and downstream amplification fragments were digested with *Kpn* I restriction enzyme, and then the two fragments were ligated with T4 DNA ligase. Using the ligation product as a template, PCR amplified with the *msmX*-P1 and *msmX*-P4 primers. The amplificon was digested with *Sac* I and *Xba* I restriction enzymes and then attached to the corresponding digestion site of the pKSV7 plasmid [43]. The recombinant plasmid was transformed into strain NCD-2 via electroporation, and *msmX* knockout was conducted via in-frame deletion, as described by Arnaud et al. [44]. The deletion of *msmX* (Δ*msmX*) was confirmed with PCR and sequencing with primers *msmX*-P1 and *msmX*-P4. To complement the Δ*msmX* mutant, intact *msmX* was amplified from strain NCD-2 with primers CP*msmX*-F (GGGGTACCTTATCGAATTCTCATTTCTG) and CP*msmX*-R (GCAGGTCGACATTGGAAATATGCACGAAAA), which included the *Kpn* I and *Sal* I restriction sites, respectively. The amplicon was digested with *Kpn* I and *Sal* I and inserted into pHY300PLK, which is a shuttle vector for *E. coli* and *B. subtilis* [45]. The recombinant plasmid was transformed into mutant strain Δ*msmX* via electroporation to obtain the complemented strain (CP*msmX*). All strains were cultured in M9 medium with L-arabinose as the sole carbon source, after which the bacterial concentration and sporulation efficiencies were calculated via plate-counting as described above.

### 4.8. Statistical Analyses

Statistically significant differences (*p* < 0.05) in NCD-2 CFU, sporulation efficiency, as well as the CFU and sporulation efficiency, between wild-type and mutant strains were evaluated with ANOVA using SPSS 18.0 software (SPSS, Chicago, IL, USA) followed by Tukey’s post hoc test. Figures were prepared with Origin Pro 8.6 software (OriginLab Corporation, Hampton, MA, USA).

## 5. Conclusions

In this study, we used PMs technology to screen several nutrients with high metabolic activity in strain NCD-2, among which L-arabinose can significantly increase the bacterial concentration and sporulation efficiency of strain NCD-2 and repress cell autolysis. The transcriptome results show that L-arabinose up-regulated the expression of sporulation-related genes and down-regulated the expression of cannibalism-related genes. Knocking out *msmX*, which is responsible for transporting arabinose, significantly decreased the bacterial concentration and sporulation efficiency of strain NCD-2 in the medium with L-arabinose as the carbon source. These results will assist in the study of directed fermentation and the mechanism of regulating the sporulation of strain NCD-2.

## Figures and Tables

**Figure 1 ijms-24-17472-f001:**
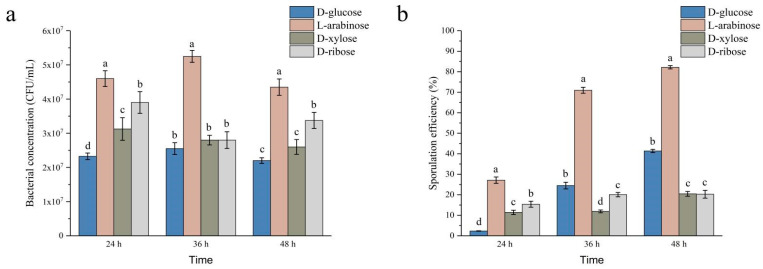
Effects of different carbon sources on the bacterial concentration (**a**) and sporulation efficiency (**b**) of strain NCD-2. Columns represent the averages of four replicates; error bars show standard deviations; and different letters indicate significant (*p* < 0.05) differences according to ANOVA with Tukey’s post hoc test.

**Figure 2 ijms-24-17472-f002:**
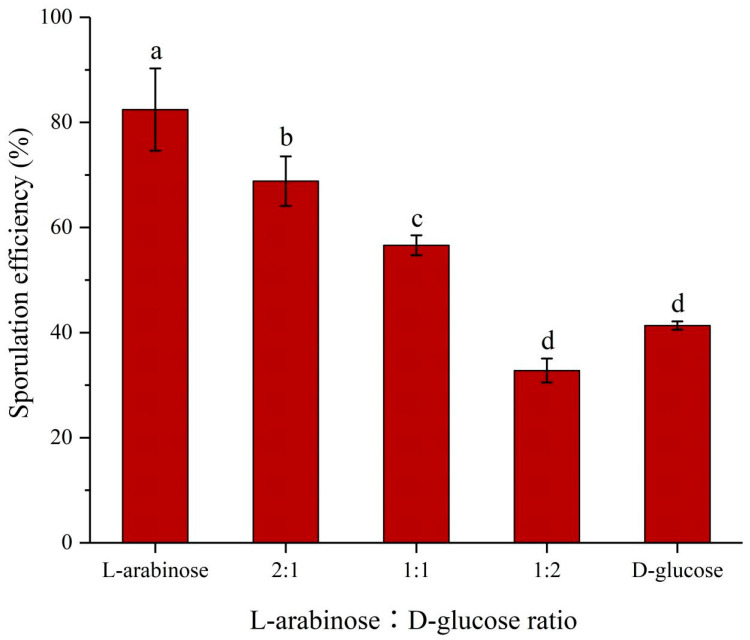
The effects of different proportions of L-arabinose and D-glucose on the sporulation of strain NCD-2. Strain NCD-2 was cultured in M9 medium containing 4 g·L^−1^ of carbohydrates, and the sporulation efficiency was determined 48 h after inoculation. Columns represent the averages of four replicates; error bars show standard deviations; and different letters indicate significant (*p* < 0.05) differences according to ANOVA with Tukey’s post hoc test.

**Figure 3 ijms-24-17472-f003:**
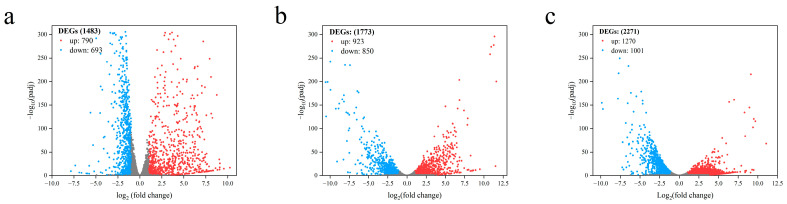
Volcano plots of transcriptomics comparison of strain NCD-2. X-axis indicates the average of log2 fold change from the replicates. Negative values indicate down-regulation, and positive values indicate up-regulation. Y-axis is -log10 padj. Dots in blue or red indicate differentially expressed genes. Dots in grey indicate proteins that are not significantly changed in gene expression. (**a**) Eight hours post-inoculation. (**b**) Twelve hours post-inoculation. (**c**) Sixteen hours post-inoculation.

**Figure 4 ijms-24-17472-f004:**
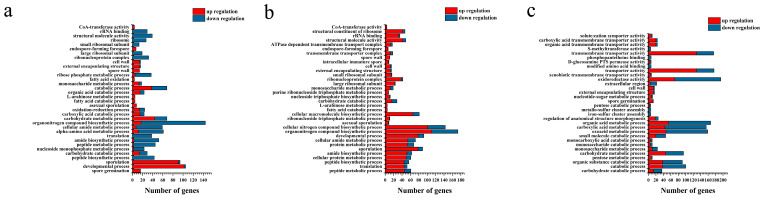
GO enrichment analysis based on the differential expression genes between L-arabinose- and D-glucose-cultured strain NCD-2. The X-axis indicates the number of genes classified into regulatory or functional categories, as depicted on the Y-axis. Columns in blue indicate down-regulation and those in red indicate up-regulation. (**a**) Eight hours post-inoculation. (**b**) Twelve hours post-inoculation. (**c**) Sixteen hours post-inoculation.

**Figure 5 ijms-24-17472-f005:**
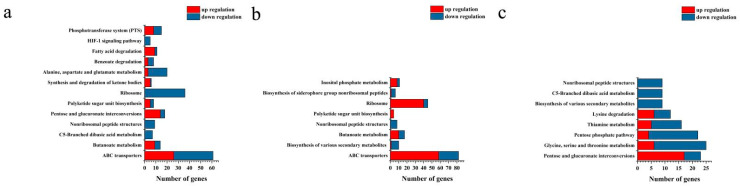
KEGG enrichment analysis based on the differential expression genes between L-arabinose and D-glucose cultured strain NCD-2. The X-axis indicates the number of genes classified into regulatory or functional categories, as depicted on the Y-axis. Columns in blue indicate down-regulation and those in red indicate up-regulation. (**a**) Eight hours post-inoculation. (**b**) Twelve hours post-inoculation. (**c**) Sixteen hours post-inoculation.

**Figure 6 ijms-24-17472-f006:**
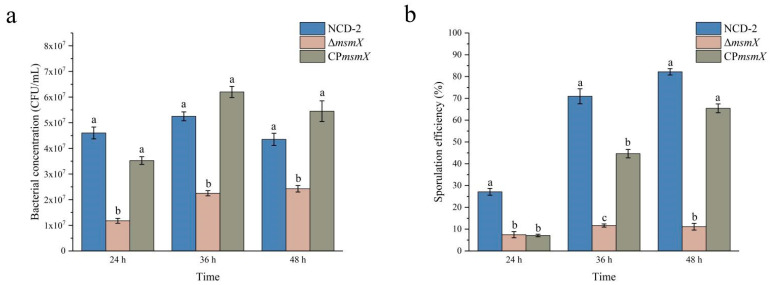
The bacterial concentration (**a**) and sporulation efficiency (**b**) of wild-type strain NCD-2 (WT), *msmX*-null mutant (Δ*msmX*), and its complemented strain (CP*msmX*) in M9 medium with L-arabinose as carbon source at 24, 36, and 48 h post-inoculation. Columns represent the averages of three replicates; error bars show standard deviations; different letters indicate significant (*p* < 0.05) differences according to ANOVA with Tukey’s post hoc test.

**Table 1 ijms-24-17472-t001:** Differentially expressed sporulation-related genes in L-arabinose-treated strain NCD-2.

Accession ID	Gene Name	Log_2_ (Ara/Glc)			Production
		8 h	12 h	16 h	
WP_003231833.1	*cotE*	7.65	5.81	2.56	Outer spore coat protein CotE
WP_003243364.1	*cotF*	6.59	9.47	5.54	Spore coat protein CotF
WP_080344234.1	*cotG*	3.55	5.59	3.82	Spore coat protein CotG
ADV92699.1	*cotM*	5.01	1.53	4.21	Spore coat protein (outer)
WP_047183078.1	*cotS*	3.98	6.01	4.53	Spore coat protein CotS
PSM02245.1	*cotT*	5.88	5.95	8.39	Spore coat protein
AGE63031.1	*cotV*	4.37	6.32	5.46	Spore coat protein (insoluble fraction)
WP_069486390.1	*cotW*	4.53	6.81	5.73	Spore coat protein
WP_014476454.1	*cotX*	4.28	6.83	6.13	Spore coat protein
WP_003231888.1	*dpaA*	6.29	6.84	4.81	Dipicolinic acid synthetase subunit A
WP_003231884.1	*dpaB*	5.76	6.12	4.21	Dipicolinate synthase subunit B
WP_015383228.1	*yheD*	5.17	3.35	2.27	Spore coat associated protein YheD
WP_063336053.1	*gerBA*	4.55	2.20	4.73	Spore germination protein GerKA
WP_003184172.1	*gerE*	3.94	5.01	3.91	Spore germination protein GerE
WP_014478336.1	*gerQ*	7.95	4.77	1.32	Spore coat protein GerQ
WP_047182746.1	*gerT*	5.10	6.02	6.03	Spore germination protein GerT
AKE24397.1	*sigK*	6.45	4.08	3.26	RNA polymerase sporulation-specific sigma factor
WP_047182864.1	*spoIIIAE*	3.84	2.72	1.41	Stage III sporulation protein AE
WP_003221804.1	*spoIIID*	10.25	6.81	4.40	Sporulation transcriptional regulator SpoIIID
WP_004398593.1	*spoIIM*	1.78	1.84	2.35	Stage II sporulation protein M
WP_047183325.1	*spoIIQ*	5.69	1.78	−1.82	Stage II sporulation protein SpoIIQ
WP_004398697.1	*spoIVB*	6.19	1.77	2.12	SpoIVB peptidase
WP_015483522.1	*spoIVFB*	1.25	1.39	1.84	Stage IV sporulation protein SpoIVFB
WP_003230465.1	*spoVAD*	6.29	1.70	4.08	Stage V sporulation protein AD
AGE63365.1	*spoVD*	2.67	3.20	2.32	Penicillin-binding protein
WP_047182441.1	*yjcA*	6.69	5.32	5.67	Sporulation protein YjcA
WP_015383520.1	*ykvU*	3.60	3.01	4.18	Sporulation protein YkvU
WP_003223491.1	*sspA*	7.22	3.14	3.04	Alpha/beta-type small acid-soluble spore protein
WP_003233287.1	*sspB*	7.39	3.43	2.66	Alpha/beta-type small acid-soluble spore protein
WP_003218568.1	*sspD*	7.57	3.93	4.00	Alpha/beta-type small acid-soluble spore protein
BAI84385.2	*sspE*	6.59	4.34	3.30	Gamma-type small acid-soluble spore protein
WP_003244950.1	*sdpC*	−1.40	−5.57	−6.81	Sporulation-delaying protein family toxin
WP_003228357.1	*sdpI*	−4.53	−3.43	−4.90	Immunity protein SdpI
WP_003243541.1	*sdpR*	−4.51	−3.34	−2.64	Sporulation-delaying system autorepressor SdpR

**Table 2 ijms-24-17472-t002:** Strains used in this study.

Strain	Genotype	Source
WT	*Bacillus subtilis* NCD-2 wild type	Lab stock
Δ*msmX*	NCD-2 mutant, *msmX*-deletion mutant	This study
CP*msmX*	Complementary of Δ*msmX* with intact *msmX*, chloramphenicol-resistant	This study

## Data Availability

Data is contained within the article and Appendix A.

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
