# Peer review of "Arabinose Plays an Important Role in Regulating the Growth and Sporulation of Bacillus subtilis NCD-2"

_ijms, 2023, doi:10.3390/ijms242417472_

Round 1

Reviewer 1 Report

Comments and Suggestions for Authors

Dear author,

 I think the idea of the paper is very perspective, to improve the sporulation of the Bacillus subtilis, and subsequently to decrease their cost for manufacturing and probably increase their usage in disease control. The topics of ecological pest and fungal control are important and should be definitely more discussed. However, the paper was hard to read and understand, and I am suggesting a complete revision of the article.

Here are some other comments:

1.       Abstract is too long (340 words), according to the MDPI guideline only 200 words are allowed. Moreover, some claims give so sense at all (probably language issue) or they negate

2.       English need urgent revision

3. The aim of the articles must be better described (in more understandable way)

4.       Please revise all the scientific names, they must be written italics (also for the supplementary material)

5.       Supplementary material: I do not understand figure s3, there is not a Venn diagram, so please explain what we see and what are axes x, y and the number, as well as the green labels. Also I do not understand the figure S1, and their role in the paper

6.       The methods are according my FAIR rules – cultivation conditions are missing (.. if I want to perform the experiment, what are ‘medium as needed and ‘appropriate concentration of antibiotics? Which antibiotics were used?). Conditions of centrifugation are missing. Manufacturer of most chemicals are missing.

7.       After the reverse transcription, it is not necessary (better to say, it is not wanted) to adjust the cDNA – if the RNA had 50 ng/ul, then we assume that it is all available RNA transcribed into cDNA. However, after measuring the cDNA again after RNA transcription, probably there will be a horribly high value (about thousand ng), which is, of course, not true because there is also primers (random primer, oligodt or other) and of course dNTPs –if from this reason, I am not trusting the real time data as presented here.

Comments on the Quality of English Language

The paper is very hard to read and understand, and I am suggesting a complete revision of the article.

Reviewer 2 Report

Comments and Suggestions for Authors

In this manuscript, the authors demonstrated that arabinose plays an important role in regulating the growth and sporulation of Bacillus subtilis NCD-2, a strain known for a promising biocontrol effect against plant soil-borne diseases and used as a commercial microbial fungicide against cotton verticillium wilt in China.

The experimental work and data collection were carried out to a high standard.

The discussion is also coherent and there is an interesting hypothesis on how arabinose can increase sporulation.

The paper is clear and interesting, although it needs to be checked by a native English speaker.

Thus, before the manuscript can be considered for publication it needs to be revised to improve clarity.

minor revisions

- Please check the English.

-   In my opinion, it is better to write bacterial growth, concentration .. and not bacteria growth ....

- in the Results is better to write  complemented strain   and not complementary  strain 

- FigS2  To see many more spores with arabinose than with glucose, it should be better to indicate the spores with an arrow

In the attached file, the sentences to be written more clearly and some suggestions are indicated in yellow

Comments on the Quality of English Language

The paper needs to be checked by a native English speaker.
